# Enhanced metabolic entanglement emerges during the evolution of an interkingdom microbial community

Giovanni Scarinci [1,2], Jan-Luca Ariens[1,2], Georgia Angelidou [1], Sebastian Schmidt[1,2], Timo Glatter [1], Nicole Paczia [1] & Victor Sourjik [1,2]

While different stages of mutualism can be observed in natural communities, the dynamics and mechanisms underlying the gradual erosion of independence of the initially autonomous organisms are not yet fully understood. In this study, by conducting the laboratory evolution on an engineered microbial community, we reproduce and molecularly track the stepwise progression towards enhanced partner entanglement. We observe that the evolution of the community both strengthens the existing metabolic interactions and leads to the emergence of de novo interdependence between partners for nitrogen metabolism, which is a common feature of natural symbiotic interactions. Selection for enhanced metabolic entanglement during the community evolution repeatedly occurred indirectly, via pleiotropies and trade-offs within cellular regulatory networks, and with no evidence of group selection. The indirect positive selection of metabolic dependencies between microbial community members, which results from the direct selection of other coupled traits in the same regulatory network, may therefore be a common but underappreciated driving force guiding the evolution of natural mutualistic communities.

Microorganisms are typically part of communities that display a large taxonomic diversity and in which members are often linked through obligatory metabolite exchanges[1–4]. These interactions likely developed through a stepwise process, resulting in a gradual erosion of independence of the initially autonomous organisms[5,6]. Notably, similar processes likely guided eukaryogenesis[7,8] and the emergence of obligate symbiotic interactions[5,6]. However, whilst communities composed of partners linked by various degrees of entanglement can be found in nature, the investigation of these evolutionary snapshots allows drawing only limited conclusions about the dynamics, molecular mechanisms and selection forces behind transitions towards increased cooperation[5,9]. Artificial synthetic communities may hence represent valuable models for the controlled observation of evolutionary processes in a relatively short time[10]. This approach was previously used to assemble mutualistic communities through the co-

cultivation of microbial partners carrying auxotrophies[11–17], some of which could be evolved towards reinforced metabolite exchanges[18–24]. Nevertheless, the next phase of the community evolution predicted by ecological models[25–27] - an increase in the interdependence mediated by the loss of traits - has not been experimentally reproduced so far. Moreover, there is still only limited validation for theoretical frameworks proposed to explain how enhanced cooperation may be evolutionarily favoured over selfish behaviours[5,6,10,11,15,25,28,29].

In this study, we aim to experimentally reproduce the transition towards increased cooperation by evolving an interkingdom mutualistic consortium between auxotrophs of *Escherichia coli* and *Saccharomyces cerevisiae* (MESCo). One advantage offered by MESCo is the ability to control the aggregation status between its two partners, since *E. coli* is naturally able to co-aggregate with yeast via *type I* fimbriae[17]. This allows us to assess the possible influence of group

[1]Max Planck Institute for Terrestrial Microbiology, Marburg, Germany. [2]Center for Synthetic Microbiology (SYNMIKRO), Marburg, Germany.
e-mail: victor.sourjik@mpi-marburg.mpg.de

selection, the major theoretical mechanism proposed to drive the emergence of cooperation[5,6,24,28]. Moreover, natural symbiotic interactions, including the most pronounced examples of metabolism reduction, often involve eukaryotic and prokaryotic organisms[6,30–33]. Compared to the previously investigated monospecies consortia between either bacteria[13,18] or yeast[14,20], the two MESCo partners display larger differences in their metabolism and exometabolome profiles[34], which is expected to favour cross-feeding interactions[35] and may allow for a greater degree of mutual de novo adaptability and metabolic specialisation of the partners. Indeed, the evolution of MESCo in our experiments leads not only to an extensive reinforcement of the initial metabolic dependencies as observed previously for purely bacterial[18,19,22] or eukaryotic[20] communities, but also to the emergence of a new interdependence in the consortium resulting from a strong reduction of ammonia assimilation by the yeast partner, and consequently its increased reliance on the bacterial exometabolome as source of nitrogen. Finally, through the use of model organisms as the community members, we perform an in-depth molecular investigation of the underlying genetic and metabolic changes.

## Results

### Experimental evolution of the MESCo communities leads to a rapid enhancement of growth

In order to identify a MESCo community that would be suitable for experimental evolution, we first investigated the ability of different auxotrophs of *E. coli* and *S. cerevisiae* to complement each other for growth in the selective cross-feeding minimal medium (CF-MM). As *E. coli* partners, we tested both fimbrialess (*ΔfimA*) strains that cannot bind to *S. cerevisiae* cells (Supplementary Fig. 1a) and fimbriated (Fim⁺) strains that can form mixed aggregates with yeast (Supplementary Fig. 1b). Despite previously reported challenges of their co-culturing[36], we recently described conditions that enable stable propagation of a synthetic community between *S. cerevisiae* and *E. coli*[17]. After 120 h of incubation, growth in selective minimal medium was observed for multiple pairs of auxotrophs (measured as optical density at 600 nm; $OD_{600}$), with aggregating communities generally reaching lower culture densities, as already reported previously[17]. We subsequently selected several cross-feeding communities, with partly overlapping partner auxotrophies and different values of $OD_{600}$ reached after 120 h, to be co-cultured over multiple (10–15) passages in CF-MM (Supplementary Fig. 1a, b). However, the majority of these tested communities turned out to be unsuitable for such long-term community propagation under conditions of cross-feeding, either due to a community collapse or because of the spontaneous regaining of prototrophy by one of the partners. An exception was the MESCo community composed of *E. coli ΔhisG* and *S. cerevisiae Δarg1* strains, which could be stably propagated as a consortium in the selective cross-feeding minimal medium (CF-MM) lacking histidine and arginine. This was true for both the non-aggregating consortia (referred to simply as MESCo) and the aggregating consortia containing fimbriated *E. coli* (referred to as MESCo^Agg).

Already after 15 regular transfers with a 1:10 dilution into fresh CF-MM, corresponding to a total of approximately 50 generations, growth of these MESCo communities greatly improved compared to the ancestral (A) community (Fig. 1a, b). These communities evolved for 50 generations under cross-feeding conditions were referred to as CF₅₀. MESCo^Agg retained their aggregative phenotype, while MESCo remained non-aggregating (Supplementary Fig. 1c), and the auxotrophic status of both partners was also retained (Supplementary Fig. 1d). The growth of both the bacterial (*Ec*^CF50) and yeast (*Sc*^CF50) partners within the evolved communities strongly improved compared to the ancestral community, and it was comparable between CF₅₀ MESCo (Fig. 1c) and MESCo^Agg (Fig. 1d), indicating that the negative effect of aggregation on growth of the ancestral community could be compensated. In our subsequent characterisation, we primarily focused on the detailed analysis of the MESCo communities evolved in the absence of aggregation.

When grown in CF-MM, communities evolved under cross-feeding (i.e., also in CF-MM) displayed both a moderate increase in the maximum growth rate ($\mu_{max}$) (Fig. 1e) and a dramatic reduction in the time to reach $\mu_{max}$ compared to the ancestral communities (Fig. 1f). In contrast, communities evolved for approximately 100 generations in the supplemented minimal medium containing both arginine and histidine (AH-MM), where growth of both partners did not require cross-feeding (referred to as S, for supplemented), showed no increase in $\mu_{max}$ and no shortening of the time to reach $\mu_{max}$ when subsequently grown in CF-MM (Fig. 1b and Supplementary Fig. 2a–c). Thus, the reduction in the time to reach $\mu_{max}$ appears to be specific for the evolution of community under conditions of cross-feeding. Consistently, there was a small but significant additional reduction in the time to reach $\mu_{max}$ in three out of four communities obtained by subsequent evolution of CF₅₀ ₃, up to a total of 269 generations (referred to as CF ₂₆₉) (Fig. 1f), whereas $\mu_{max}$ and the final cell densities only increased for one of the CF ₂₆₉ communities (Fig. 1b, e). Notably, the ratio between both partners remained relatively stable over the course of evolution (Supplementary Fig. 2d–g).

When the ancestral and evolved communities were co-cultured in either CF-MM or AH-MM, both evolved *E. coli* (*Ec*^CF) and *S. cerevisiae* (*Sc*^CF) strongly outcompeted the ancestral strains (*Ec*^A and *Sc*^A) in CF-MM, while being outcompeted in AH-MM (Fig. 1g, h). This confirms that the improved growth of the evolved MESCo communities in CF-MM is due to a specific advantage under conditions of cross-feeding. In the absence of metabolic interdependency, this adaptation rather imposes a fitness cost, which might explain why it did not occur during the evolution in the supplemented medium.

### A small set of mutations captures genetic changes in evolved communities

Sequencing the genomes of evolved populations of *E. coli* and *S. cerevisiae* revealed a small set of common mutations, appearing largely in the same sequential order in communities evolved without or with aggregation (Fig. 2a and Supplementary Data 1-4). The first mutations that became fixed in all evolved communities interrupted *argR*, which encodes the transcription factor that is known to repress the biosynthesis and transport of arginine and the transport of histidine[37], and to regulate several other metabolic pathways in *E. coli* (Fig. 2b). Whilst inactivation of ArgR may enhance cross-feeding due to the increased production of arginine, such overproduction could also impose a metabolic burden. Indeed, the introduction of *ΔargR* (*Ec*^ΔR) mutation into the ancestral *E. coli* strain led to a decreased growth rate (Fig. 2c), but higher levels of arginine (Fig. 2d) in the histidine-supplemented minimal medium. The increased excretion of arginine, required by the yeast partner, could explain the improvement in growth of the evolved communities, but given its negative effect on *E. coli* growth, the apparent strong positive selection on *argR* inactivation was counterintuitive. However, based on a previous study[37], these mutations could also derepress the *hisJQMP* operon encoding the histidine transporter (Fig. 2b), which was confirmed by the elevated activity of the *hisJ* promoter in *Ec*^ΔR strain (Fig. 2e). Thus, the partner-serving inactivation of *argR* in *E. coli* is likely selected indirectly, due to an increased histidine uptake rather than enhanced cross-feeding. Consistent with positive selection on the histidine uptake, another set of *E. coli* mutations (in 4 out of 8 lines after 50 generations, and in all lines after 269 generations) directly affected the *hisJ* promoter region, resulting in a further increase in the promoter activity additionally to that provided by the *argR* deletion (Fig. 2e and Supplementary Fig. 3a). Notably, no mutations in these genes were observed in *E. coli* strains evolved in the supplemented medium (AH-MM).

The selective advantage of the identified mutations was confirmed by co-culturing different *E. coli* strains under cross-feeding conditions in the presence of the ancestral yeast strain (*Sc*^A) (Fig. 2f).

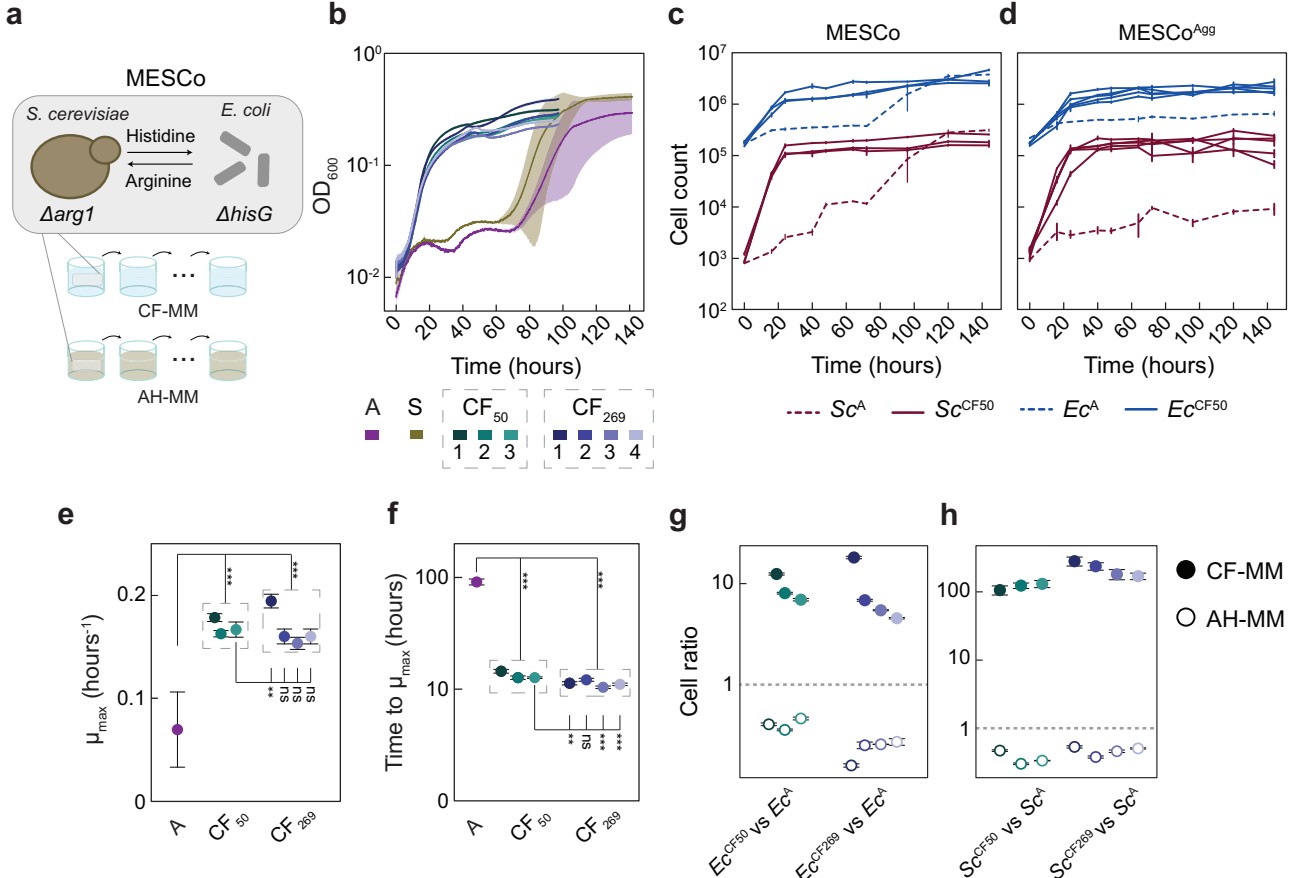

**Fig. 1 | Design and experimental evolution of the MESCo communities.**
**a** Laboratory evolution of the MESCo communities in minimal media, either selective for cross-feeding (CF-MM) or supplemented with arginine and histidine (AH-MM). **b** Growth in CF-MM measured as optical density at 600 nm ($OD_{600}$) of the non-aggregating ancestral community (A) and of communities evolved either in CF-MM, for 50 ($CF_{50}$) or 269 ($CF_{269}$) generations, or in AH-MM for 100 generations (S). Colours and numbers denote different lines of evolution, and solid lines and shading indicate the mean values calculated from 3 biological replicates ($n$) and the corresponding standard deviation (SD). **c** Counts of *E. coli* (blue) and *S. cerevisiae* (red) cells in non-aggregating ancestral (dashed lines) or evolved (solid lines) communities during growth in CF-MM. **d** Same counts for aggregating communities. Cell counts were measured here and throughout using flow cytometry. Mean values of $n = 3$ biological replicates ± SD are shown. **e** Maximum growth rate and **f** time required to reach it for growth curves shown in **b**. Mean values of $n = 3$

biological replicates ± SD are shown. Significance comparisons were performed between the ancestral and either all $CF_{50}$ or all $CF_{269}$ communities, and between $CF_{50\,3}$ community and all $CF_{269}$ communities that were derived from it. Cell ratio of the evolved bacterial ($Ec^{CF}$; **g**) or yeast ($Sc^{CF}$; **h**) lines upon competition between evolved and ancestral community members, labelled with different fluorescent markers, which were inoculated in a co-culture at equal optical densities, and grown for 96 h either in CF-MM or in AH-MM as indicated. Cell ratio was calculated as the ratio of the final cell counts for the indicated competing strains, normalised by their ratio at inoculation. A value of 1 represents absence of growth bias between the competing strains whilst values higher or lower than 1 indicate that the first strain respectively outcompetes or is outcompeted by the second one. Mean values of $n = 5$ biological replicates ± SD are shown. $p$ values (ns = $p > 0.05$, **$p < 0.01$, ***$p < 0.001$) reported in **e** and **f** are from a one-way ANOVA followed by Tukey post-hoc test. Source data are provided as a Source Data file.

The ancestral *E. coli* strain ($Ec^A$) was outcompeted by $Ec^{\Delta R}$, and even stronger by a strain carrying both the $\Delta argR$ and a mutation in the *hisJ* promoter ($Ec^{\Delta RH+}$), while $Ec^{\Delta R}$ was outcompeted by the $Ec^{\Delta RH+}$ strain. A similar result was obtained when these *E. coli* strains were all co-cultured together with $Sc^A$ under cross-feeding conditions, with the single mutants outcompeting the ancestral strain but being out-competed by the double mutant strain (Supplementary Fig. 3b). Consistent with our mutation analysis, $Ec^{\Delta RH+}$ strain largely recapitu-lated the phenotype of the evolved *E. coli*, being only slightly out-competed by the $Ec^{CF269}$ lines in the co-culture (Fig. 2f).

A similarly small set of common mutations was present at high frequencies in the evolved lines of *S. cerevisiae* (Fig. 2a), along with a number of low-frequency mutations (Supplementary Data 2). One prominent group of mutations introduced a premature stop codon in the gene *ecm21*, also known as *art2*. Ecm21 is a positive regulator of ubiquitination of several amino acid transporters, including the arginine transporter Can1, which promotes their endocytosis and subsequent degradation[38] (Fig. 2g). Inactivation of Ecm21 likely benefits yeast under cross-feeding

conditions because of the increased cell-surface levels of transporters and therefore increased uptake of amino acids, including arginine required by yeast. Notably, the emergence of similar mutations in *ecm21* was previously reported after the co-evolution between two different yeast auxotrophs[20]. Selection for the increased levels of Can1 may also explain the amplification of the entire chromosome V or of its region encoding the *can1* gene (Supplementary Fig. 4).

Another group of mutations in all the evolved yeast lines inter-rupted *gdh1*, a gene encoding glutamate dehydrogenase. Gdh1 is the primary glutamate dehydrogenase used by *S. cerevisiae* growing on glucose[39], and it catalyses one of the two major reactions for assimilation of ammonium (Fig. 2g). Although surprising given the presence of ammonium in the growth medium, the apparent selection for the loss of ammonium assimilation during evolution is further supported by the emergence of nonsense mutations (in 3 lines after 50 genera-tions, and in all, except one, lines after 269 generations) in *glt1*, a gene that encodes the enzyme catalysing the second branch of direct ammonium assimilation (Fig. 2a, g).

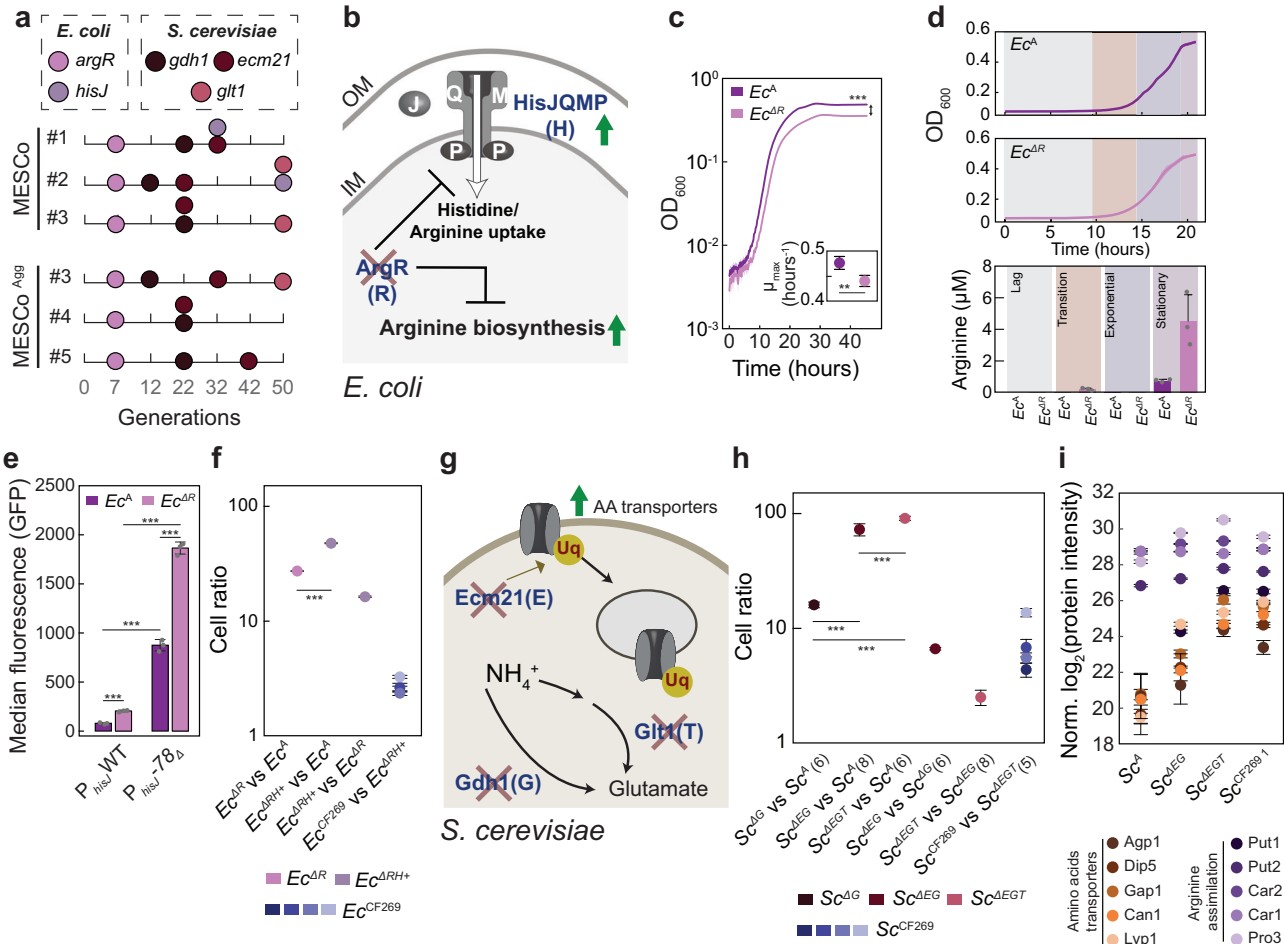

**Fig. 2 | Common mutations fixed within the evolved MESCo communities.**
**a** Identified high-frequency mutations and time of their first detection for communities evolved without (MESCo) or with (MESCo^Agg) aggregation. Line labelling as in Fig. 1. **b** Functions of affected proteins and expected impact on *E. coli* (cross: nonsense mutations; up arrow: increased expression). **c** Growth, and maximum growth rate (*Inset*), of the ancestral (*Ec*^A) and and *ΔargR* (*Ec*^ΔR) *E. coli* in CF-MM supplemented with histidine. *n* = 5 biological replicates. **d** Extracellular arginine measured at different growth phases for *Ec*^A or *Ec*^ΔR cultures grown as in **c**. *n* = 3 biological replicates. **e** Activity of the reporter plasmid, carrying either the wildtype (WT) *hisJ* promoter or one of its mutated versions (deletion in the nucleotide in position -78 from the start codon of *hisJ*; *-78Δ*) in front of the *gfp* gene, measured in either *Ec*^A or *Ec*^ΔR background. *n* = 3 biological replicates. **f** Cell ratios of the indicated *E. coli* strains (*ΔRH* + : *ΔargR* and chromosomal $P_{hisJ}$ *-78Δ*) in pairwise competition in CF-MM, co-cultured with the ancestral yeast partner calculated as the ratio of the final cell counts for the indicated competing strains, normalised by their ratio at inoculation. *n* = 4 (*Ec*^ΔR, *Ec*^ΔRH+) or 5 (*Ec*^CF269) biological replicates.
**g** Functions of affected proteins and possible impact on *S. cerevisiae* partner (cross: nonsense mutations). **h** Relative growth of the indicated *Sc* strains (A: ancestral; *ΔG*: *Δgdh1*; *ΔEG*: *Δecm21 Δgdh1*; *ΔEGT*: *Δecm21 Δgdh1 Δglt1*) in pairwise competition in CF-MM, co-cultured with *Ec*^ΔR as a partner. The number of biological replicates is shown in the brackets for each comparison. **i** Normalised intensity of indicated proteins measured in indicated yeast strains co-cultured in CF-MM for 36 h with *Ec*^ΔR. *n* = 4 biological replicates. *p* values are reported in (Supplementary Table 1). *P* values (ns = *p* > 0.05, \**p* < 0.05, \*\**p* < 0.01, \*\*\**p* < 0.001) reported in **c**, **e** and **f** are from a two-tailed *t*-test assuming unequal variance between the samples, in **h** and **i** are from a one-way ANOVA followed by Tukey post-hoc test. Mean values ± SD are shown throughout. Source data are provided as a Source Data file.

When introduced in the ancestral yeast lines, these common mutations again conferred cumulative fitness advantage under cross-feeding conditions when introduced in the order of their appearance in the ancestral yeast strain, evidenced by co-incubation of different mutant yeast strains with *Ec*^ΔR (Fig. 2h). This *E. coli* strain was chosen as a partner because *argR* mutations appeared in the community prior to any yeast mutations. Notably, there was a gradual decrease of the relative benefit for fitness provided by each subsequent mutation, with the deletion of *gdh1* giving the strongest benefit, followed by *ecm21* and then by *glt1*, which likely explains their order of fixation in the population. As for *E. coli*, these common mutations apparently capture most, but not all, of the beneficial changes in the evolved *S. cerevisiae* populations, since the *Sc*^CF269 lines were moderately fitter than the triple knockout (*Sc*^ΔEGT) strain (Fig. 2h).

The proteomics analysis confirmed that the evolved communities and the community formed by the engineered *Sc*^ΔEGT and *Ec*^ΔRH+ mutant strains exhibit largely similar changes in protein levels compared to the ancestral community. These included upregulation of the HisJQMP transporter and proteins involved in the uptake and biosynthesis of arginine in *E. coli* (Supplementary Fig. 5a and Supplementary Data 5). Besides these common changes, both evolved lines, *Ec*^CF269 1 and *Ec*^CF269 2, showed a downregulation of the histidine biosynthetic pathway, as well as an upregulation of the outer membrane porin OmpF (Supplementary Fig. 5b–d and Supplementary Data 5) that is consistent with the mutations in the *ompF* promoter in these *Ec*^CF269 lines (Supplementary Data 1). Similarly, changes in the *S. cerevisiae* proteome were largely overlapping between the evolved lines and the mutant community, including the expected upregulation of the arginine transporter Can1 (Supplementary Data 6). However, the interpretation of these data was complicated by the difference in growth between the ancestral and evolved or mutant communities. We therefore analysed the proteome of different *S. cerevisiae* strains (either evolved or the engineered *Sc*^ΔEGT mutant) co-cultured

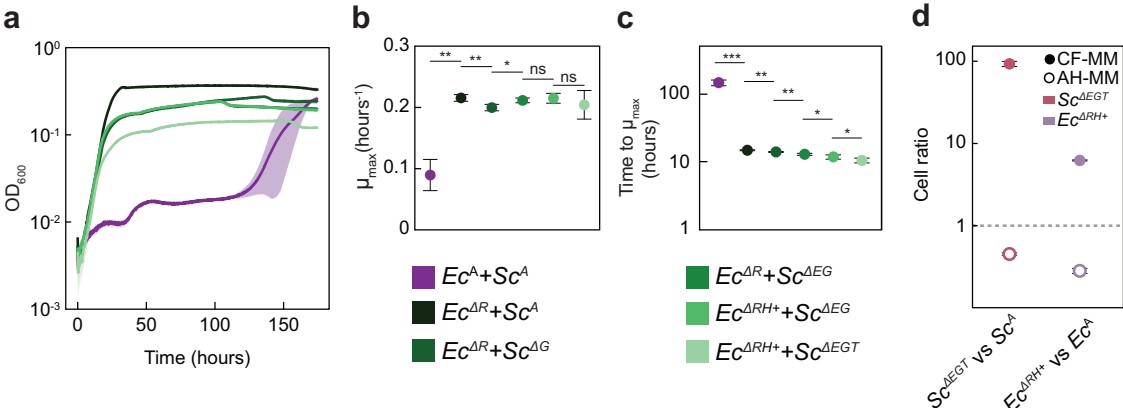

**Fig. 3 | Impact of mutations on growth of MESCo communities. a** Growth in CF-MM of co-cultures between indicated mutant strains representing consecutive stages of the community evolution. **b** Corresponding maximal growth rate and **c** time to reach it. Mean values of $n = 4$ biological replicates ± SD are shown. **d** Cell ratio of yeast and bacterial mutants carrying the main mutations observed during evolution, in direct competition either in CF-MM or AH-MM with the ancestral counterparts. Mean values of $n = 5$ biological replicates ± SD are shown. $p$ values (ns = $p > 0.05$, *$p < 0.05$, **$p < 0.01$, ***$p < 0.001$) in **b**, **c** are from a one-tailed $t$-test assuming equal variances between samples. Source data are provided as a Source Data file.

with the same $Ec^{\Delta R}$ partner, to ensure similar growth of all tested communities. These results confirmed elevated levels of Can1 in the $Sc^{CF269\,1}$ line and in the strains carrying key mutations, and further demonstrated the upregulation of several other amino acid transporters as well as proteins involved in arginine assimilation and nitrogen metabolism (Fig. 2i, Supplementary Fig. 6a, b and Supplementary Data 7).

In order to better understand the sequence in which mutations were fixed (Fig. 2a), we reconstructed communities between individual mutants. The inactivation of *argR* produced the most pronounced effect on growth (Fig. 3a), reflected in the strongly increased growth rate (Fig. 3b) and an even more dramatic reduction in the time to reach $\mu_{max}$ (Fig. 3c). Thus, the major enhancement of the community growth in our experiments occurs rapidly, in less than seven generations by which time *argR* mutations consistently appeared in all *E. coli* lines, and it is apparently caused by the increased production and release of arginine by the *argR* mutant that abolishes the initial growth delay observed in the ancestral community. The effects of all subsequent mutations were less strong, with no or even negative impact on the growth rate or final density of the communities, but with a gradual reduction in the time to reach $\mu_{max}$. Given that a similar reduction was observed in the evolved communities (Fig. 1b, f), the time to reach $\mu_{max}$ may be the main feature under evolutionary selection. The reconstituted communities carrying the major mutations were further able to phenotypically mimic the evolved lines (Supplementary Fig. 7a-f), including the ability to outcompete the ancestral community members under cross-feeding conditions, whilst being outcompeted in the supplemented medium (Fig. 3d and Supplementary Fig. 7g, h).

We also compared the growth of communities containing only individual mutations. While the deletions of *argR* or *ecm21* improved the growth of the community, the deletion of *gdh1* and the mutation of the *hisJ* promoter led to a reduction or cessation of the community growth (Supplementary Fig. 7i). The ability of these mutations to collapse the community growth when combined with the ancestral partner likely explains why they could only be fixed at subsequent stages of the community evolution, after inactivation of *argR*. Particularly, the apparent dependence of *S. cerevisiae gdh1* mutant growth on *E. coli argR* mutation supports the notion of the co-evolution between the community members.

### Evolved yeast strains have a strongly reduced ability to directly assimilate ammonium

Whereas mutations related to the uptake and/or biosynthesis of histidine and arginine could enhance the pre-existing metabolic

interactions within the MESCo communities, the fixation of yeast mutations affecting ammonium assimilation was unexpected. Nevertheless, further mutations in genes related to ammonium assimilation were observed in $Sc^{CF269}$ lines (Fig. 4a and Supplementary Data 2), which include mutations in *gdh3* (paralogue of *gdh1*), in *mep1* and *mep3* that encode ammonium transporters, and premature stop codons in *gln3* that encodes the global transcriptional regulator of the nitrogen metabolism. Truncation of Gln3 was reported to produce a constitutively active version of this regulator[40], and upregulation of Gln3 targets (including Can1) was indeed observed in the $Sc^{CF269\,2}$ yeast line carrying such truncation, when compared to the $Sc^{CF269\,1}$ line that retained the intact form of *gln3* (Supplementary Fig. 8a). The reduced ability of the evolved lines, as well as the corresponding yeast knock-out strains, to assimilate ammonium was confirmed by growing them in arginine-supplemented minimal medium, either with or without ammonium. While the ancestral yeast strain exhibited much better growth in the presence of ammonium, the benefit from ammonium assimilation was largely reduced for both the evolved and mutant strains (Fig. 4b–d and Supplementary Fig. 8b, c), and such reduction was only marginal for control strains evolved in AH-MM without cross-feeding (Supplementary Fig. 8d, e).

When their preference for the source of nitrogen was directly tested, by growing yeast cells in the presence of the isotope-labelled ammonium and unlabelled arginine (Fig. 4e), the fraction of [15]N-labelled proteinogenic amino acids, except arginine, was indeed much higher in the ancestral yeast strain compared to the evolved lines or to the mutant strains (Fig. 4f and Supplementary Fig. 9a). The effect was even more pronounced for the evolved lines compared to the $Sc^{\Delta EGT}$ strain, likely because of the aforementioned additional mutations (Fig. 4a and Supplementary Data 2). The deletion of *gdh3* and *gdh2* in the $Sc^{\Delta EGT}$ strain (labelled as $Sc^{5KO}$) indeed resulted in a further reduction in the amino acid labelling (Supplementary Fig. 9a). Consistent with their increased reliance on arginine, the concentration of arginine in the supernatant of either evolved or mutant yeast strains was much lower compared to the ancestral yeast strain (Fig. 4g), implying an increase in consumption of arginine per $OD_{600}$ unit (Fig. 4h).

We also tested the production of histidine, the metabolite provided by yeast to the bacterial partner within the consortium. However, in contrast to the general upregulation of arginine biosynthesis by the *E. coli* partner, only the $Sc^{CF269\,1}$ line, originating from the fastest-growing evolved community (Fig. 1e), showed increased abundance of histidine, and several other amino acids, in the supernatant (Fig. 4i and

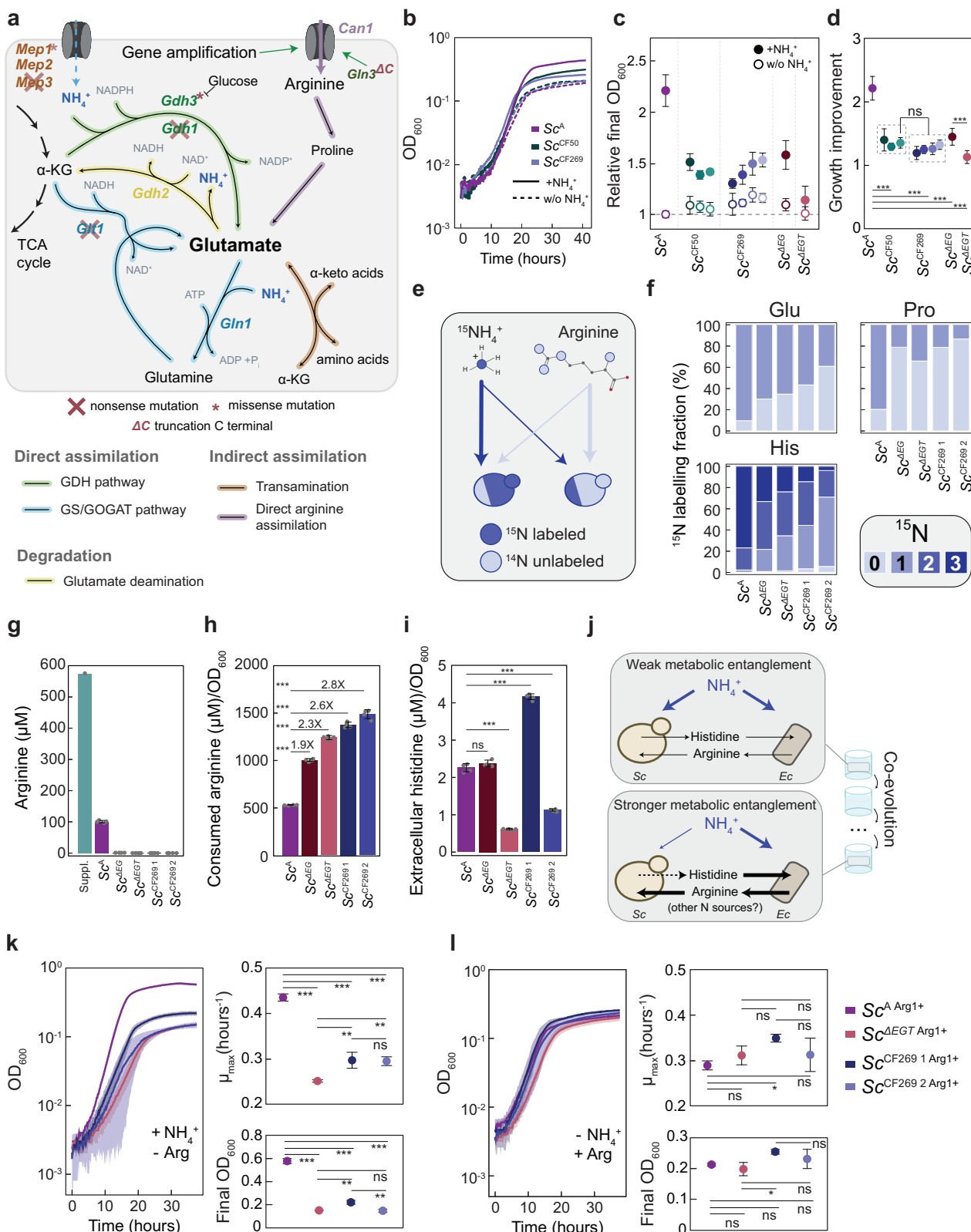

Supplementary Fig. 9b). The $Sc^{\Delta E}$ mutant instead showed an unchanged level, and the $Sc^{\Delta EGT}$ mutant and the $Sc^{CF269\ 2}$ line showed even decreased levels of histidine, in contrast to the previous observation that *ecm21* mutations increased metabolite sharing in a yeast cross-feeding community[20]. Thus, the $Sc^{CF269\ 1}$ line may have acquired additional mutation(s) that, possibly together with the *ecm21* mutation, enabled it to increase the cross-feeding of its partner.

We thus conclude that the evolution under cross-feeding conditions led to the increased reliance of yeast on arginine, supplied by its bacterial partner, as the primary nitrogen source instead of ammonium (Fig. 4j). This occurred despite the fact that ammonium was present in the medium during the entire course of evolution. To further verify this conclusion, we decoupled the use of arginine as the nitrogen source from the arginine auxotrophy in the yeast strains, by

**Fig. 4 | Modified nitrogen source preference of the evolved yeast partner.**
**a** Ammonium and arginine uptake and assimilation pathways in *S. cerevisiae*, showing mutations detected in the evolved yeast lines (cross: nonsense mutations; asterisk: missense mutations, ΔC: truncation of Gln3). **b** Representative growth curves showing impact of addition of ammonium on growth of the indicated yeast lines in the arginine-supplemented minimal medium. $n = 1$. **c** Final $OD_{600}$ relative to the ancestral strain grown in the absence of ammonium, and **d** growth increase due to the presence of ammonium for indicated yeast strains. A, $n = 11$; $Sc^{\Delta EG}$, $Sc^{\Delta EGT}$, $n = 9$; $Sc^{CF50}$, $Sc^{CF269\,1}$, $n = 5$; $Sc^{CF269\,2-4}$, $n = 8$ biological replicates. **e** Illustration of the labelling experiment and the expected difference in labelling. **f** Average fraction of $^{15}N$-labelling in amino acids from cultures in stationary phase. $n = 4$ biological replicates, with SD (not shown) below 3% for all samples. **g** Concentration of arginine present in the supplemented CF-MM medium at inoculation (Suppl) and in the spent media of indicated yeast strains and **h** calculated consumption of arginine

per unit of $OD_{600}$. $n = 4$ biological replicates. **i** Histidine levels per unit of $OD_{600}$ detected in the spent media of indicated *S. cerevisiae* strains. $n = 4$ biological replicates. **j** Changes throughout evolution in MESCo communities, with cross-feeding between partners highlighted in black and ammonium assimilation in blue. Thicker black arrows indicate an enhancement in uptake and/or release. The dashed black arrow indicates variable enhancement of histidine release among yeast lines. The thinner blue line indicates reduced assimilation of ammonium. **k** Growth, maximal growth rate and final $OD_{600}$ of indicated strains with restored arginine prototrophy in minimal media with ammonium as the primary nitrogen source. **l** Same parameters for growth with arginine as the primary nitrogen source. $n = 3$ biological replicates in **k** and **l**. $p$ values (ns = $p > 0.05$, *$p < 0.05$, **$p < 0.01$, ***$p < 0.001$) reported in **d**, **h**, **l**, **k**, **l** are from a one-way ANOVA followed by Tukey post-hoc test. Mean values ± SD are shown throughout. Source data are provided as a Source Data file.

reintroducing the *arg1* gene to restore their arginine prototrophy. Although these "restored" strains could all grow in CF-MM without arginine, their growth on ammonium as the primary source of nitrogen was strongly reduced compared to the arginine-prototroph ancestral strain (Fig. 4k). In contrast, they grew equally well or even faster when arginine was provided instead of ammonium as the primary nitrogen source (Fig. 4l), or during residual growth on other supplements present in CF-MM (Supplementary Fig. 10).

### Restoration of prototrophy does not abolish the dependency of evolved yeast on the bacterial partner

Strains restored for the respective prototrophies further enabled us to directly test possible evolved dependencies within the community, beyond reliance on the exchange of arginine and histidine, by comparing growth of the prototroph strains in the absence and presence of the respective partner. Although cell counts of arginine prototrophs originating from the evolved yeast lines were lower compared to the restored ancestral strain (Fig. 5a), consistent with their reduced growth on ammonium (Fig. 4k), these strains grew significantly better in CF-MM in presence of $Ec^{\Delta RH+}$ (Fig. 5a, b) (one-way ANOVA $p = 0.001$, $R^2 = 0.48$). Such enhancement was not observed for the restored ancestral strain or for the restored $Sc^{\Delta EGT}$ prototroph, and it was weak for the restored 5KO prototroph, suggesting that *S. cerevisiae* indeed evolved additional dependency on the *E. coli* partner that may go beyond its increased reliance on arginine as the source of nitrogen. Supporting that, the final cell counts of the yeast prototroph strains did not differ significantly between communities containing either the ancestral or evolved *E. coli* (Fig. 5c), despite different levels of arginine excretion between these *E. coli* strains. Conversely, within the same community the evolved *E. coli* auxotroph partner does benefit more from the presence of the yeast prototroph (Fig. 5d), possibly due to its enhanced ability to scavenge histidine.

In contrast, when prototrophy was restored in *E. coli* strains, no significant difference in growth improvement was observed between the restored ancestral strain, the evolved lines and the $Ec^{\Delta RH+}$ mutant (Fig. 5e, f) (one-way ANOVA $p = 0.28$, $R^2 = 0.23$), Furthermore, *E. coli* growth was not different when the partner yeast was either the ancestral or the evolved strain, and the evolved yeast did not benefit more strongly from the bacterial prototroph compared to the ancestral yeast strain (Fig. 5g, h). Thus, whereas yeast evolved additional dependencies on the bacterial partner beyond the originally engineered mutualism, *E. coli* enhanced its ability to profit from histidine but not from other metabolites provided by the yeast partner.

## Discussion

Ecological models predict that enhanced partner addiction should emerge from the co-evolution of interdependent organisms[27,41], such as those exchanging essential metabolites in mutualistic and symbiotic communities[2–4,32]. In this study, we report experimental observation of an increase in metabolic entanglement during the experimental

laboratory evolution of an engineered interkingdom mutualistic community between auxotrophs of *E. coli* and *S. cerevisiae* (MESCo).

Using this synthetic interkingdom community formed by two model organisms enabled us to mechanistically describe several characteristic steps in the progression of communities towards tighter cooperation. This evolution firstly included the strengthening of pre-existing interactions, through the self-serving enhanced uptake of the exchanged metabolites by both partners, as previously observed in bacterial or yeast communities[16,20,21]. The cooperation was further promoted by the costly increase in sharing the partner-serving metabolite by *E. coli*, and at least in one instance also by yeast. Previous observations of the enhanced excretion of the partner-serving metabolites were primarily made in the context of laboratory evolution of spatially structured communities[18,19,24], and the formation of multicellular clusters was even favoured by the co-evolution[23]. These instances of selection on cooperative traits could thus be interpreted as a consequence of local cooperation within small neighbourhoods[42], favouring group selection that is normally assumed to be a prerequisite for the evolution of cooperation[5,6,10,11,15,28].

In contrast, although group selection was specifically enabled in our experiments in the MESCo^Agg community, the cooperative metabolite sharing rather emerged as a consequence of the pleiotropic mutations in the same regulatory component that simultaneously increased production of the partner-serving metabolite and uptake of the self-serving metabolite. While it has previously been demonstrated that pleiotropy can stabilise existing cooperation against the emergence of cheaters through regulatory coupling between cooperative and private traits[43–45], the relevance of pleiotropy in the evolution of social traits has been questioned[29,46]. Nonetheless, at least in one case pleiotropy was proposed to explain the selection of the *ecm21* mutations in a mutualistic yeast community[20], which may also apply to our experiments where the same yeast gene was mutated. Repeated instances of the apparently indirect selection on increased interdependency between partners during the short evolution of our MESCo communities suggest that pleiotropy may be a generally important, and previously underappreciated, factor in the evolution of sociality, promoting the emergence of social traits. Our results also indicate a mechanism that could favour selection of such pleiotropic over purely self-serving mutations, because of the observed negative impact of the latter on community growth.

Besides reinforcements of the pre-existing interactions, the evolved MESCo communities showed repeated emergence of a new level of dependency, with the yeast partner becoming increasingly reliant on *E. coli* for assimilation of ammonium, the primary nitrogen source in the medium during the co-culture evolution. This increased entanglement evolved through sequential inactivation of the major pathways of ammonium assimilation in yeast. Importantly, fixation of the most prominent of these mutations, *gdh1*, was apparently contingent on the prior fixation of the *argR* mutation in *E. coli*, indicative of co-evolution between partners. Although the underlying selection

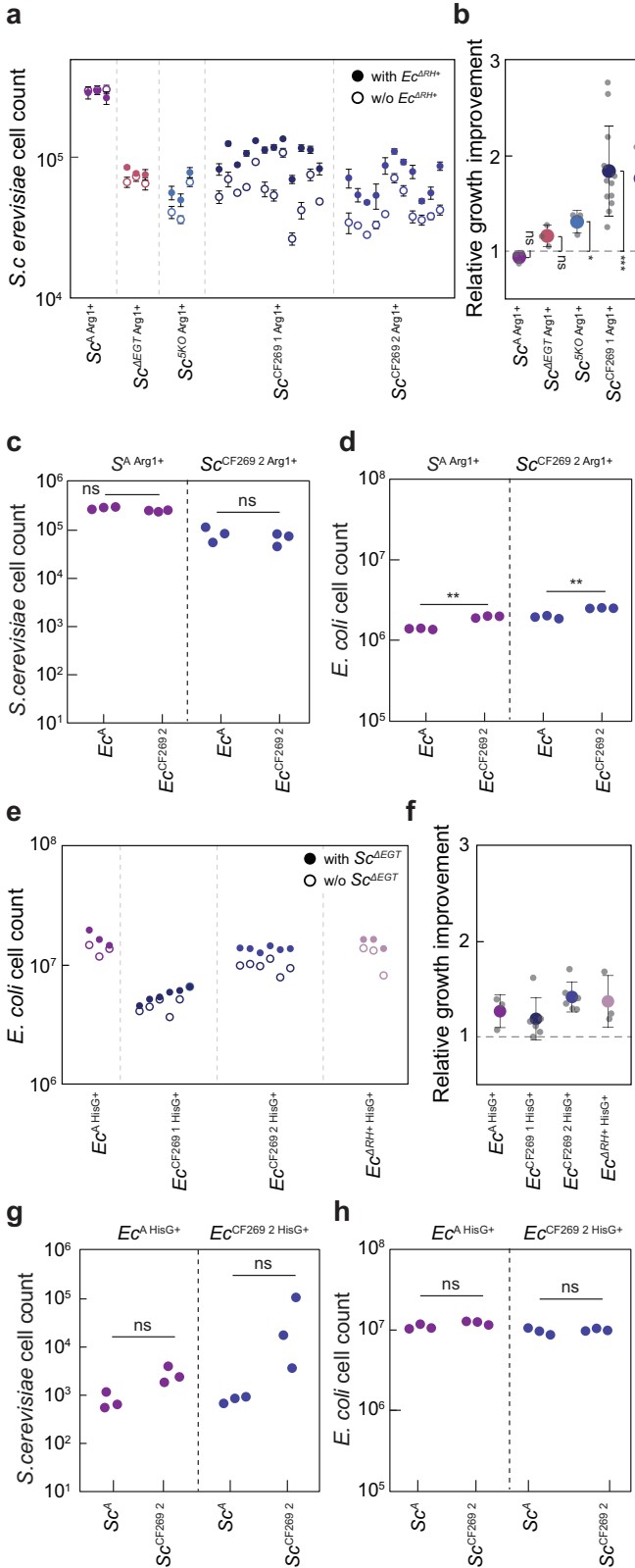

**Fig. 5 | Partner dependencies of the evolved community members after restoration of their prototrophy. a** Growth of indicated *S. cerevisiae* strains with restored arginine prototrophy in the presence or absence of $Ec^{\Delta RH+}$ in CF-MM for 80 h, shown as mean values ($n = 3$) $\pm$ SD of the resulting cell counts. **b** The corresponding relative growth improvement in the presence of *E. coli*, calculated as the ratio between cell counts with and without bacterial partner. Counts of **c** yeast and **d** bacterial cells in the co-cultures between indicated restored *S. cerevisiae* prototroph strains and either ancestral or evolved ($Ec^{269\,2}$) *E. coli* auxotrophs, grown as in **a**. Each dot in all panels represents the cell count from individual transformants and is calculated as the average of two independent cultures. $Sc^{A}$, $Sc^{\Delta EGT}$, $Sc^{5KO}$, $n = 3$; $Sc^{CF269\,1}$, $n = 12$; $Sc^{CF269\,1}$, $n = 11$ biological replicates. **e** The growth of indicated *E. coli* strains with restored histidine prototrophy in presence or absence of $Sc^{\Delta EGT}$ in CF-MM for 80 h and **f** the relative growth improvement calculated as the ratio between cell counts with and without the yeast partner. Each dot represents the average of two biological replicates. $Ec^{A}$, $Ec^{\Delta RH+}$, $n = 3$; $Ec^{CF269}$, $n = 6$ biological replicates. Counts of **g** yeast and **h** bacterial cells in the co-cultures between indicated restored *E. coli* prototroph strains and either ancestral or evolved ($Sc^{269\,2}$) *S. cerevisiae* auxotrophs, grown in CF-MM for 80 h. Each dot represents the cell count from individual transformants and is calculated as the average from two biological replicates. *p* values (ns = $p > 0.05$, **$p < 0.01$, ***$p < 0.001$) reported in **b** are from a one-sample two-tailed *t*-test assessing for a difference from a value of 1 while in **c, d, g, h** are from a two-tailed *t*-test assuming unequal variance between the samples. Source data are provided as a Source Data file.

example in the redox potential[48], which could select for its inactivation. In either case, we conclude that this erosion of autonomy is again selected indirectly, as a consequence of regulatory trade-offs within the yeast metabolic network.

In summary, we observed that high regulatory connectivity of the cellular genetic and metabolic networks leads to the fixation of mutations that enhance existing and create novel metabolic interdependencies in the evolving synthetic microbial community, even in the absence of group selection. We speculate that similar mechanisms may drive the emergent division of labour during the evolution of natural communities, including other dependencies based on shared nitrogen-containing compounds that are common in symbiotic interactions[32,49,50].

## Methods

### Strain construction

*E. coli* knockout strains interrupted in different metabolic pathways were obtained from the Keio collection[51] (Supplementary Data 8). Additional gene deletions in *E. coli* were introduced using pSIJ8 (Supplementary Table 2) as described[52], after rescuing the strain from the kanamycin resistance (*kanR*) as described in[53]. Cassettes containing the *kanR* resistance cassette were amplified using respective gene deletion strains from the Keio collection, with flanking homology regions of 100 to 150 bp. For the introduction of the point mutations in the *hisJ* promoter region, cassettes containing the *neo-ccdB* were amplified from pKD45 using the primers GS_288 and GS_289 (Supplementary Table 3), which include a 50-bp homology from both sides for the desired region. This cassette was introduced in the desired *E. coli* strain as described[53], and strains carrying this cassette were then transformed with a fragment containing the desired region amplified from the evolved lines, followed by selection on M9 minimal media plates containing rhamnose as sole carbon source. To distinguish strains during competition experiment using specific fluorescent markers, *E. coli* strains were transformed with pNB1[54] and pOB2[55] (for two-strain competition) or with pGS62-65 (for four-strain competition) (Supplementary Table 2).

*S. cerevisiae* strains were obtained from the respective knockout collection[56] (Supplementary Data 9). Additional deletions in *S. cerevisiae* were obtained by transforming the desired strains with cassettes obtained by PCR amplification of the hygromycin B resistance from pH3FS[57] (Supplementary Table 2), with flanking homology regions of

pressure remains to be fully elucidated, the reduced ability of yeast to use ammonium may cause a rewiring of the nitrogen assimilatory pathways to enhance the uptake and consumption of arginine, thereby providing mutants with an increased scavenging ability for this metabolite, and thus with a competitive fitness advantage in a cross-feeding community. Additionally, the assimilation of ammonium under conditions of cross-feeding may cause some metabolic imbalance[47], for

50 bp targeting the desired locus. Positive candidates were then transformed with the Cre-containing plasmid pPL5071[58] (Supplementary Table 2) and positive colonies were selected on minimal media plates lacking uracil and further screened via PCR to confirm the correct removal of the cassette containing the antibiotic resistance. Subsequently, after an overnight growth of positive candidates on complete minimal media at 30 °C, a selection on minimal media plates containing 5-fluoroorotic acid was performed to select for colonies lacking the pPL5071 plasmid. To distinguish strains during competition experiment via fluorescent markers, cassettes containing the *mNeonGreen* and *mTurquoise2* genes were amplified respectively from pMFM073[59] or pGS5 (Supplementary Table 2) and integrated in the *his3* locus of *S. cerevisiae*. For each competition experiment presented, the first strain listed is labelled with mTurquoise2 while the second always with mNeonGreen. For all the other experiments, strains expressing mTurquoise2 were used.

## Growth conditions
For *E. coli* pre-cultures, cells were inoculated directly from glycerol stocks into 5 ml lysogeny broth (LB), and if required the appropriate antibiotic was added. Pre-cultures were incubated at 37 °C for 16–18 h with shaking at 200 r.p.m. For *S. cerevisiae* pre-cultures, strains were firstly streaked from glycerol stocks on yeast extract peptone dextrose (YPD) plates, supplemented, when necessary, with the appropriate antibiotic. After incubation for 48 h at 30 °C, six colonies were inoculated in 5 mL YPD, supplemented, when necessary, with the appropriate antibiotic. Pre-cultures were incubated at 30 °C for 16–18 h with shaking at 200 r.p.m. Cells from 2 ml of the pre-cultures were collected by centrifugation, washed twice with phosphate saline buffer (PBS), suspended in 1 ml PBS and incubated for 5 h, either at 30 °C (*S. cerevisiae*) or 37 °C (*E. coli*). Cross-feeding experiments were performed in the low fluorescence (LoFlo) yeast nitrogen base (YNB) minimal medium (Formedium Ltd) buffered with 100 mM 2-(N-morpholino)ethanesulfonic acid (MES) (Roth) at pH 6.15, containing 2 % D-glucose as the carbon source and a mixture of 100 mg/l L-leucine, 20 mg/l L-methionine and 20 mg/l uracil to complement the auxotrophies present in the *S. cerevisiae* strains in the knockout collection[56] (cross-feeding medium; CF-MM). Unless otherwise stated, both organisms were inoculated at $OD_{600}$ of 0.05 each, values referring to a 1-cm cuvette. For competition experiment, each strain tested for each organism was inoculated in equal amount to a final total initial $OD_{600}$ of 0.05. When supplements were used, a concentration of 100 mg/l of arginine and/or 20 mg/l of histidine were added to the minimal medium (supplemented minimal medium; AH-MM). Growth was measured as $OD_{600}$ in a plate reader (Infinite 200 Pro, Tecan), by inoculating either the monocultures or the co-cultures in 48-well plates in 300 μl of the desired media and incubating the cultures for indicated time at 30 °C with 200 r.p.m. shaking. Growth plots were generated using the seaborn package[60] in Python and quantification of growth parameters was done using QurvE[61] (non-parametric model). Statistical analyses were performed with GraphPad Prism v9.0.2.

## Experimental evolution
For the evolutionary experiments, co-cultures were inoculated in 1 ml of either CF-MM or AH-MM supplemented with 50 μg/ml kanamycin in 24-well plates and grown at 30 °C with 200 r.p.m shaking. For the evolution in CF-MM, the co-cultures were transferred into fresh medium at a ratio 1:10 (with ~3.3 generation per passage) every seven days for the first fifteen transfers (approximately 50 generations), at a ratio of 1:10 every 3.5 days between the 16th and the 35th transfer, and at a ratio of 1:200 until the 55th transfer (for a total of approximately 269 generations). For lines evolved in AH-MM, co-cultures were transferred fourteen times at a ratio of 1:100 every 24 h (approximately 100 generations). In order to isolate different organisms from the co-cultures, communities were streaked respectively on YPD supplemented with

50 μg/ml of streptomycin to isolate yeast, and on LB supplemented with 50 μg/ml of nystatin to select for *E. coli*. The majority of colonies present on the plates were then pooled and grown in liquid cultures in the respective selective rich media, and these cultures were used to prepare glycerol stocks.

## Aggregation assay and microscopy
The aggregation assay was performed as described before[17]. Bacterial and yeast cells from pre-cultures, grown as described above, were washed twice with PBS and mixed together in 1 ml PBS in a 24-well plate (Greiner Bio-One GmbH) at a final $OD_{600}$ of 0.7 for *S. cerevisiae* and 0.2 for *E. coli*. Plates were then incubated with shaking at 200 r.p.m. for 1 h at 30 °C and imaged using a Nikon SMZ745T stereo microscope.

## Sequencing
For Sanger sequencing, the genomic region of interest were firstly amplified by PCR (Q5-NEB), and the products were purified using the DNA Clean & Concentrator kit (Zymo Research). For the next-generation sequencing (NGS) of genomes of populations of individual partners, genomic DNA extractions were performed using the NucleoSpin Microbial DNA Mini kit (Macherey-Nagel) following manufacturer's instructions. In brief, pellets from 2 ml of LB-grown overnight cultures of *E. coli* were resuspended in 2 ml of the lysis buffer and homogenised (2 x 20 s at 6800 r.p.m. using Precellys Evolution, Bertin Technologies SAS). Pellets from 2 ml of YPD-grown overnight cultures of *S. cerevisiae* were resuspended in the lysis buffer, transferred to 400 μl suspension of HCl-treated glass beads (Merck KgAA) and vortexed with a Vortex Genie 2 (neoLab Migge GmbH) for 5 min at maximum speed. DNA concentration was quantified using a Qubit 4 Fluorometer (Thermo Fisher Scientific). For sequencing of $CF_{50}$ communities, libraries were prepared using the Nextera XT DNA Library Preparation Kit (Illumina), and then sequenced using a Miniseq (Illumina). For $CF_{269}$ communities, both library generation (NGS DNA Library Prep set-Novogene) and sequencing (Illumina NovaSeq 6000 S4 flowcell- Illumina) were performed by Novogene Co. Analysis of the sequencing data was performed using breseq[62] and Integrative Genomics Viewer (IGV- version 2.8.9)[63] and data are presented in (Supplementary Data 3, 4). Original fasta sequencing data are deposited in NCBI under the Bioproject PRJNA1049669 for $CF_{50}$ and under the Bioproject PRJNA1051099 for $CF_{269}$.

## Construction and analysis of promoter reporters
Plasmids carrying the *gfp* reporter under the control of different versions of *hisJ* promoter were constructed using the NEBuilder HiFi DNA Assembly (NEB). Primers used to amplify the backbone[64] or the promoter regions from the evolved bacterial lines are reported in Supplementary Table 3. *E. coli* strains transformed with the reporter plasmid were grown in 500 μl CF-MM supplemented with 20 mg/l histidine in a 48-well plate for 45 h at 30 °C, and afterwards fluorescence was measured via flow cytometry. In this case, the $Ec^A$ rescued from *kanR* resistance was used as ancestral strain.

## Flow cytometry analysis
Flow cytometry measurements were performed using the BD LSR Fortessa SORP cell analyzer (BD Biosciences). A 488-nm laser line, with a power set to 20%, was used to determine both side scatter (SSC) and forward scatter (FSC) values, and combined with a 510/20 BP filter to detect GFP fluorescence. A 447-nm laser line combined with a 470/15BP filter was used to detect mTurquoise2 fluorescence, while the same laser line combined with a 586/15 BP was used to detect lss-mOrange[65]. mCherry fluorescence was measured using a 561-nm laser line combined with 632/22 BP filter. *S. cerevisiae* and *E. coli* populations were distinguished by FSC and SSC, and the respective different strains used for competitions experiments were distinguished according to their respective fluorescent labelling (mCherry, GFP, mNeonGreen,

mTurquoise2 and lss-mOrange for *E. coli*, mNeonGreen, mTurquoise2 for *S. cerevisiae*). Measurements were performed using the BD High Throughput Sampler (HTS) with a fixed flow rate set at 0.5 μl/s for an acquisition time of 20 s, with samples diluted in PBS to yield of $10^3$–$10^4$ cell counts per second. If necessary, *S. cerevisiae-E. coli* aggregates were disrupted as described previously[17], by diluting the community in PBS supplemented with 4% mannose followed by pipetting. The abundance of cells in the defined volume (10 μl) was inferred from the sample dilution, flow rate and sampling time. Flow cytometry data were analyzed using FlowJo (BD Biosciences).

### Proteomics sample preparation and liquid chromatography-mass spectrometry (LC-MS) measurements

To facilitate the collection and preparation of samples for the proteomic analysis, incubation was performed in trans-wells, where *S. cerevisiae* and *E. coli* partners are separated by a membrane (0.4 μm, Cellquart), which allows the metabolite exchange. Specifically, 3 ml of CF-MM containing *E. coli* partner at an initial $OD_{600}$ of 0.083 were transferred into each well of a 6-well plate (SARSTEDT AG & Co. KG), a trans-well was inserted, and 2 ml of CF-MM containing *S. cerevisiae* partner at an initial $OD_{600}$ of 0.125 was added. Cultures were grown at 30 °C with shaking at 110 r.p.m.

Cells (equivalent to a total $OD_{600}$ of 3.0) were harvested and washed three times with ice-cold PBS (15,000 g, 10 min, 4 °C) and resuspended in 300 μl of the lysis buffer containing 2% sodium lauroyl sarcosinate (SLS) and 100 mM ammonium bicarbonate. *E. coli* samples were then heated for 10 min at 90 °C, while *S. cerevisiae* samples were heated for 90 min at 90 °C. Samples were then ultra-sonicated for 10 s at maximum power (Vial Tweeter, Hielscher). Proteins were reduced with 5 mM tris (2-carboxyethyl) phosphine (Thermo Fisher Scientific) at 90 °C for 15 min and alkylated using 10 mM iodoacetamid (Sigma Aldrich) at 20 °C for 30 min in the dark. After centrifugation for 10 min at 13,000 g, supernatants were transferred into a new tube. For *S. cerevisiae*, extracts were acetone-precipitated with a four-fold excess of ice-cold acetone and incubation for 18 h at −20 °C, washed twice with methanol and dried for 10 min at room temperature. Dry pellets were then reconstituted in 200 μl lysis buffer. For both organisms, the amount of proteins was determined by bicinchoninic acid protein assay (Thermo Fisher Scientific).

50 μg of extracted protein were incubated with 1 μg of trypsin (SERVA Electrophoresis GmbH) and 0.5% SLS at 30 °C overnight. Following digestion, the peptide samples was acidified using trifluoroacetic acid (TFA) (Thermo Fisher Scientific) to precipitate SLS. C18 solid phase extraction (SPE) cartridges (Macherey-Nagel) were used to desalt and concentrate peptides. SPE cartridges were prepared by adding acetonitrile (ACN), and equilibrated with 0.1% TFA before sample load. After peptide binding, cartridges were washed with aqueous buffer containing 5% ACN and 0.1% TFA. Peptides were eluted with 50% ACN and 0.1% TFA, dried and reconstituted in 0.1% trifluoroacetic acid.

Peptides were analyzed using liquid-chromatography-mass spectrometry with an Exploris 480 mas spectrometer connected to an Ultimate 3000 RSLC nano (both Thermo Fisher Scientific). Peptides were separated on a capillary column (75 μm x 42 cm) packed in-house with C18 resin (2.4 μm; Dr. A. Maisch HPLC GmbH). A flow rate of 300 nl/min and separating gradient from 94% solvent A (0.15% formic acid) and 6% solvent B (99.85% acetonitrile, 0.15% formic acid) to 35% solvent B over 60 min was used.

MS Acquisition was performed in DIA (data independent acquisition) mode as described previously[66]. The spray voltage was set to 2.3 kV, funnel RF level to 40, and heated capillary temperature to 275 °C. Survey scan MS resolutions for DIA experiments were set to 120.000 at m/z 200 with an AGC (Automatic Gain Control) target of 300%, a max. injection time of 50 ms and mass range of 350–1400. For DIA fragment spectra acquisition, AGC target value was set at 3000% and 45 14 Da windows were used with an overlap of 1 Da. Stepped HCD collision

energy of 25, 27.5, 30% was used. Fragment spectra resolution was set to 15,000 and max. injection time to 22 ms. MS1 data was acquired in profile mode, MS2 DIA data in centroid mode.

DIA-NN version 1.8[67] together with Uniprot databases for *E. coli* or *S. cerevisiae* was used to generate a spectral library for the DIA analysis. The neural-network based DIA-NN suite performed noise interference correction (mass correction, RT prediction and precursor/fragment co-elution correlation) and peptide precursor signal extraction of the DIA-NN raw data. The following parameters were used: full tryptic digest allowing two missed cleavage sites with oxidised methionines and carbamidomethylated cysteins as modifications. "Remove likely interferences" and "match between run" options were enabled. The neural network classifier was set to the "single-pass mode", protein inference based on "genes" and the quantification strategy was set to "any LC (high accuracy)". "RT-dependent" was used for cross-run normalisation and "smart profiling" for library generation. DIA-NN outputs were statistically evaluated using the SafeQuant[68,69] R-script modified for processing DIA-NN outputs. The SafeQuant script was executed on the "report.tsv" file from DIA-NN analysis to sum precursor intensities to represent protein intensities. The peptide-to-protein assignment was done in SafeQuant with redundant peptide assignment following the Occam´s razor approach. Median protein intensity normalisation was performed followed by imputation of missing values using a normal distribution function. Log-ratio and significance value (Student´s *t*-Test) calculation was performed as a basis for volcano plots with Perseus[70]. The mass spectrometry proteomics data have been deposited to the ProteomeXchange Consortium via the PRIDE[71] partner repository with the dataset identifier PXD047443. Protein association network analysis and functional enrichment were performed with STRING[72].

### Sample preparation for metabolite quantification

In order to quantify the arginine concentration in the *E. coli* supernatants, the cultures of the bacterial strains grown in 48 wells plates containing 300 μl of CF-MM supplemented with 20 mg/l histidine were filtered through a 15 mm 0.2 μm pore size reconstitute cellulose filters (Phenomenex Ltd) and flow-through samples were stored at −80 °C until measurement without any further treatment. In this case, the $Ec^A$ rescued from the *kan* resistance gene was used as ancestral strain. For *S. cerevisiae* metabolites measurement in presence of $^{15}N$ ammonium, cells were inoculated with an initial OD of 0.01 in 24-well plates containing 1500 ml of CF-MM without ammonium, supplemented with 100 mg/l arginine and 5g/l 98 % $^{15}N$ $(NH4)_2SO_4$ (Merck KgAA), and grown at 30 °C until stationary phase. For metabolite quantification from supernatants, cultures were filtered and stored as above. For proteinogenic amino acid hydrolysis and extraction, samples were adjusted to equal biomass according to $OD_{600}$, cells were collected by gentle centrifugation (3000 g, 10 min), and the pellets were washed three times with PBS. Washed pellets were suspended in 6N HCl solution and transferred to glass vials with conical base (ROTILABO-Carl Roth) and stored at 98 °C for 6 h. Samples were then dried under a nitrogen stream, suspended in 250 μl double distilled water and transferred into clean Eppendorf tubes. These were centrifuged at maximum speed for 10 min and the supernatants were transferred into clean Eppendorf tubes and stored at -80 °C until measurement without any further treatment.

### Metabolite quantification via LC-MS

Both quantitative and qualitative determination of the target metabolites were performed using HRES LC-MS. The chromatographic separation was performed on a Vanquish HPLC System (Thermo Fisher Scientific) using a ZicHILIC SeQuant column (150 × 2.1 mm, 3.5 μm particle size, 100 Å pore size) connected to a ZicHILIC guard column (20 × 2.1 mm, 5 μm particle size) (Merck KgAA), with a constant flow rate of 0.3 ml/min. The temperature was maintained at 25 °C. The two

mobile phases were a solution of 0.1 % Formic acid in 99:1 water: acetonitrile (Honeywell research chemicals) as mobile phase A, and a solution of 0.1 % formic acid 99:1 acetonitrile:water (Honeywell research chemicals) as phase B. The injection volume used per each sample was set to 5 μl. The following steps and linear gradients were used for the mobile phase profile: 0 – 8 min from 80 to 60 % B; 8 – 10 min from 60 to 10 % B; 10 – 12 min constant at 10 % B; 12 – 12.1 min from 10 to 80 % B; 12.1 to 18 min constant at 80 % B. ID-X Orbitrap mass spectrometer (Thermo Fisher Scientific) was used in positive mode with a high-temperature electrospray ionisation source and the following conditions: H-ESI spray voltage at 3500 V, sheath gas at 50 arbitrary units, auxiliary gas at 10 arbitrary units, sweep gas at 1 arbitrary units, ion transfer tube temperature at 350 °C, vaporiser temperature at 350 °C. Detection was performed in full scan mode using the orbitrap mass analyser at a mass resolution of 60 000 in the mass range 50–250 (m/z). Extracted ion chromatograms of the $[M+H]+$ forms were integrated using Tracefinder 5.2 software (Thermo Fisher Scientific). For the reported intensity levels of the different amino acids, values were obtained by summing the area under the peaks from LC-MS measurements for the different isotopologues. Absolute concentrations were then calculated based on external calibration curves.

### Reporting summary

Further information on research design is available in the Nature Portfolio Reporting Summary linked to this article.

## Data availability

Original proteomics and sequencing data have been deposited in public repositories as indicated in Materials and Methods. All materials are available from the corresponding author upon request. The proteomics data have been deposited to the ProteomeXchange Consortium via the PRIDE partner repository with the dataset identifier PXD047443. Original fasta sequencing data are deposited in NCBI under the Bioproject PRJNA1049669 for $CF_{50}$ and under the Bioproject PRJNA1051099 for $CF_{269}$. Source data are provided with this paper.

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

## Acknowledgements

We thank Jörg Kahnt for the support with the proteomics analysis, Silvia Gonzalez Sierra for the support with the flow cytometry, and Elif Elçin and Paushali Chaudhury for the support with the NGS sequencing. We thank Julian Pietsch for insightful discussions. We thank John S. Parkinson for providing the materials and the protocol for gene replacement in *E. coli*. This research was funded by the Max-Planck-Gesellschaft.

## Author contributions

G.S. and V.S. conceived and designed the study. G.S., J.L.A., S.S., T.G., and N.P. performed the experiments. G.S., G.A., T.G. and N.P. analysed the data. G.S. and V.S. wrote the manuscript.

## Funding

## Competing interests

The authors declare no competing interests.

## Additional information

**Supplementary information** The online version contains
supplementary material available at

Victor Sourjik.

**Peer review information** *Nature Communications* thanks Daniele Daffonchio, who co-reviewed with Ramona Marasco and the other,
anonymous, reviewer(s) for their contribution to the peer review of this
work. A peer review file is available.

**Publisher's note** Springer Nature remains neutral with regard to
jurisdictional claims in published maps and institutional affiliations.

