## [Peer Review File · Nature Communications]

Enhanced metabolic entanglement emerges during the evolution of an interkingdom microbial communityReviewers' Comments:

Reviewer #1:

Remarks to the Author:

The manuscript by Scarinci and colleagues addresses an important: how microbial species become metabolically dependent during evolution. Addressing this topic has been a huge focal point in the field and the authors do a good job of tackling these questions through a good combination of experimental evolution of model interkingdom communities composed of *Escherichia coli* and *Saccharomyces cerevisiae*. The paper is relatively well-written, the topic is interesting and leverages the power of simple and synthetic communities to molecularly track the progression of species towards increased partner entanglement. The highlight of this paper is mechanistically tracking the molecular dynamics of the communities over evolutionary time and the clear demonstration of partner entanglement, at least for one of the entities in the community.

That said, I think the manuscript can be further improved if the authors can address the following points.

Line 49: The authors hypothesize that interkingdom microbial communities consisting of prokaryotes and eukaryotes, where partners share no natural co-evolutionary history, might be more likely to undergo an evolutionary metabolic specialisation. However, one can also argue that such interactions may fail to establish given that evolutionarily distinct microbes might be more likely to display antagonistic interactions between each other. In fact, the authors do state (lines 60-62) that a majority of pairs that they tested were unsuitable for the long-term. I suggest the authors adjust their statement.

Line 73: The phrase aggregation status suggests that both MESCo and MESCoAgg were aggregating. However, I think the authors probably mean they retain their ancestral identities. They should rephrase to make this point clear.

Figure 1: I found the multiple plots and panels, especially the statistical notations to be very crowded and confusing to decipher. In my opinion, the authors should simplify this plot and statistical annotations.

Line 87: I think the authors should not use co-evolution (at-least at this stage) as a term to reason for the effect of reduced times to reach max growth rate. I think co-existence is a better term. Co-evolution requires complementarity at the genomic levels as well. This probably comes later in the paper.

Lines 140-156: The feeling I get from this section is that the authors reconstructed mutations in the ancestral *E.coli* strains but not in the yeast strains. This is surprising since yeast is also a model system. But probably the authors have a good reason for this. It would be great if they elaborate/mention this in the paper. This would be key for showing coevolution. If a knockout mutant in yeast leads to collapse of the community than this would be a clear example of co-evolution.

Generally, I feel the manuscript misses a crucial control. Are the authors certain that the mutations in E.coli that enable it to profit from histidine, would not have occurred if they had simply evolved in a medium containing histidine. I think the authors should tone down the use of terms like co-evolution. For showing, co-evolution, I think it is imperative to show reciprocal evolutionary change in both partners that would otherwise not happen.

Reviewer #2:

Remarks to the Author:

This manuscript presents a detailed set up of an interkingdom, simplified community made of a bacterium and a yeast, to study the evolution of the metabolic interdependency of nitrogen use. By studying the different generations of the two cross-feeding members of the co-culture, the study reports that several mutations in the two members of the community are selected to enhance the nutritional cooperation and interdependency. The study relies on a large set of interdisciplinary approaches from mutant constructions to genome analyses, proteomics, metabolomics and community manipulations providing a rich set of data. The study produces a large set of evidence that demonstrates the enhancement of the metabolic interdependency with a stronger metabolic “entanglement” between the two community components. It highlights the importance of “indirect selection” in reshaping such metabolic entanglement. The study is very interesting and well-designed providing very robust proofs of an enhanced cooperation.

Two aspects should require attention: i) the level of novelty related to previous studies; ii) the connection between the introduction and the discussion of the results, regarding the robustness and rationale of the hypothesis at the base of the study. Regarding point i) It would be useful to read in the discussion an assessment of the result in comparison to previous studies that may help to highlight the novelty and importance of the study; also some discussion on the relevance of the findings for interactions in real world communities would enrich the manuscript. As per point ii) in the introduction the hypothesis proposed points out that interkingdom communities should evolve metabolic specialization more likely than communities of just bacteria or microbial eukaryotes. Besides the fact that a robust rationale on which this hypothesis is built is not clearly presented in the introduction, such hypothesis is not then developed in the discussion that touches subjects that are interesting, but not specifically related to the raised hypothesis.

Specific comments

1. Lines 12-13: The information value of this sentence is questionable. The importance of metabolic interactions in the selection processes in general are very well demonstrated and, as well, their importance in the formation of complex life forms is self-evident. It is suggested to rephrase or remove the sentence. The following sentence slightly rephrased may be an incipit of the abstract that is clearer than the current one: “Different stages of mutualism can be observed in nature, however, the dynamics and mechanisms underlying.....”.
2. Lines 20-23: These lines are difficult to read, because imply definitions not available in an abstract. The exact meaning of “indirect selection” is elusive in the context of the present text

(especially in relation to pleiotropy).

3. Line 38: “passive metabolic interactions” should be briefly explained/defined.

4. Line 48-52: From the context the current hypothesis is hardly to be considered as such, because it coincides with the observation presented at lines 45-46 that “the most pronounced examples of metabolism reduction are observed for symbiotic interactions between prokaryotic and eukaryotic partners”. A hypothesis should be different from an observation that can support it. The new concept within a hypothesis should be generated by the assembly of different solid arguments. This is an essential aspect of the study, that is the hypothesis tested should be solidly justified. In the current presentation it is not indicated the rationale of why interkingdom microbial communities (prokaryote with eukaryote) should more likely undergo a metabolic specialization, than communities of solely bacteria or microbial eukaryotes.

5. Introduction: The previous comment highlights a general problem of the introduction that does not sufficiently explain the context and the rationale of the study (and the hypothesis).

6. Extended Data Fig 1: What the reason of the choice of the co-cultures maintained for multiple (10-15) passages? If the growth is measured just with the OD, how can be measured the stability of auxotrophy?

7. Lines 62-63: The community combination indicated here is stated to be propagated stably but on panel a of extended data Fig 1 the OD seems very low.

8. Line 71 and Figure 1: This is not very clear and may confuse the reader. For instance, in Fig 1b S is the ancestral co-culture grown in AH-MM or is after 100 generations as it is apparent from the text in line 85? If this is the case the text and Fig 1b and the figure caption should be consistent, all indicating that S is after 100 generations in non-cross-feeding medium (AH-MM). Another point: No differences are observed between 50 and 269 generations, raising the question when (around which generation) the growth changed? In the ancestral culture(s) apparently there are two growth phases (diauxia/crabtree effect?) that are not present in the CF (evolved) cultures. How have been considered any eventual effect of yeast-produced ethanol on E.coli? In the figure caption panels (h,i) are g,h in the figure.

9. Line 76: What exactly mean “nearly proportional”?

10. Figure 2 and lines 103-119: The text and figure here should benefit of a revision for clarity. One difficulty for the reader is to match the time and generations. In the figure panel a), time of mutation appearance is presented to illustrate the mutation, while in the text (e.g. line 116 for E.coli, but the same for the yeast lines reported later at lines 145-146 for example) the number of generations is used making difficult to follow where the number of generations as well the number of lines (“...4 out of 8 lines after 50 generations...”) are presented.

11. Line 111-112: Is this an original hypothesis presented here or it is an observation made previously for instance in reference 35?

12. Lines 113-115: This exemplify well the concept of indirect selection that was introduced in the introductory section of the manuscript. For non-expert readers it would be useful to get such kind of clarification already in the introductory part.

13. Figure 4J: In this panel it should be useful that the meaning of the different type of arrows (thickness, dashed lines, colours, etc) may be explained in the caption.

14. Discussion: The current discussion is presenting several interesting aspects, but is not well linked to the introduction and does not provide a conceptual continuity to the hypothesis raised in the introductory part of the manuscript. It is suggested to revise the discussion/introduction to

make them more coherent and with a direct evident conceptual linkage.

Reviewer #3:

Remarks to the Author:

Point-by-point response to the reviewer comments

Reviewer #1 (Remarks to the Author):

The manuscript by Scarinci and colleagues addresses an important: how microbial species become metabolically dependent during evolution. Addressing this topic has been a huge focal point in the field and the authors do a good job of tackling these questions through a good combination of experimental evolution of model interkingdom communities composed of *Escherichia coli* and *Saccharomyces cerevisiae*. The paper is relatively well-written, the topic is interesting and leverages the power of simple and synthetic communities to molecularly track the progression of species towards increased partner entanglement. The highlight of this paper is mechanistically tracking the molecular dynamics of the communities over evolutionary time and the clear demonstration of partner entanglement, at least for one of the entities in the community.

That said, I think the manuscript can be further improved if the authors can address the following points.

We thank the Reviewer for this accurate summary of our work including its most important points. We also thank the Reviewer for the positive comments on our work and for helpful suggestions on improving the clarity of data presentation and discussion.

Line 49: The authors hypothesize that interkingdom microbial communities consisting of prokaryotes and eukaryotes, where partners share no natural co-evolutionary history, might be more likely to undergo an evolutionary metabolic specialization. However, one can also argue that such interactions may fail to establish given that evolutionarily distinct microbes might be more likely to display antagonistic interactions between each other. In fact, the authors do state (lines 60-62) that a majority of pairs that they tested were unsuitable for the long-term. I suggest the authors adjust their statement.

We apologize for unintentionally overemphasizing this particular hypothesis by stating it at the end of the introduction. Although we find this hypothesis plausible, we did not specifically aim to test it in our study (see also our response to the comment by the Reviewer #2). As suggested by both Reviewer #1 and Reviewer #2, we now rephrased the second part of the introduction to better formulate the aims of our investigation and to emphasize the advantages of our synthetic model to achieve these aims, and we rephrased this particular statement.

Line 73: The phrase aggregation status suggests that both MESCo and MESCoAgg were aggregating. However, I think the authors probably mean they retain their ancestral identities. They should rephrase to make this point clear.

We thank the Reviewer for pointing out this ambiguity. The Reviewer is correct, and this section has now been modified accordingly to improve its clarity.

Figure 1: I found the multiple plots and panels, especially the statistical notations to be very crowded

and confusing to decipher. In my opinion, the authors should simplify this plot and statistical annotations.

We apologize for the complexity of the data presentation in this figure. Statistical annotation in panels e and f has been adjusted to increase clarity.

Line 87: I think the authors should not use co-evolution (at-least at this stage) as a term to reason for the effect of reduced times to reach max growth rate. I think co-existence is a better term. Co-evolution requires complementarity at the genomic levels as well. This probably comes later in the paper.

While we believe that our data altogether support the hypothesis of a co-evolution between the two organisms, we agree with the Reviewer that the use of the term "co-evolution" might be premature at this stage of the manuscript and therefore say instead "evolution of community under conditions of cross-feeding".

Lines 140-156: The feeling I get from this section is that the authors reconstructed mutations in the ancestral *E. coli* strains but not in the yeast strains. This is surprising since yeast is also a model system. But probably the authors have a good reason for this. It would be great if they elaborate/mention this in the paper. This would be key for showing coevolution. If a knockout mutant in yeast leads to collapse of the community than this would be a clear example of co-evolution.

We apologize for not explaining it better in the text, but we did introduce the main mutations detected in the evolved yeast lines in the ancestral strain and studied their phenotype. We now rephrased this section to highlight this.

However, it is also true that when co-cultured with the ancestral *E. coli* strain, the most frequent and first mutation observed in yeast, *gdh1*, leads to the collapse of the community (as shown in Extended Data Fig. 7i). Instead, strains carrying this mutation had to be co-cultured with *E. coli argR*, the first mutation that became fixed in all lines during the community evolution. Thus, the selection for *gdh1* in the yeast partner requires prior fixation of *argR* in the bacterial partner. We now emphasize that this observation supports the hypothesis of co-evolution, and we thank the Reviewer for this suggestion.

Generally, I feel the manuscript misses a crucial control. Are the authors certain that the mutations in *E. coli* that enable it to profit from histidine, would not have occurred if they had simply evolved in a medium containing histidine. I think the authors should tone down the use of terms like co-evolution. For showing, co-evolution, I think it is imperative to show reciprocal evolutionary change in both partners that would otherwise not happen.

The control suggested by the Reviewer, evolution in a medium containing histidine, was already essentially performed and shown in Fig. 1b and Extended Data Fig.2a-c, but it was not well explained in the previous version of our manuscript (as also pointed out by the Reviewer #2). In brief, we evolved the co-culture in presence of both supplements in the arginine and histidine minimal medium (AH-MM), thus allowing growth of the two community members independent of cross-feeding. These communities did not show any major improvement in growth when subsequently tested under conditions of cross-feeding (Fig. 1b and Extended Data Fig.2a-c). We thus concluded that they are unlikely to have acquired

mutations that would enhance cross-feeding. We have now additionally confirmed that by amplifying and sequencing of *argR* and *hisJ* genes from these lines, which showed no mutations in these genes.

Moreover, the yeast lines isolated from these AH-MM-evolved co-cultures show only a marginal reduction in ammonia assimilation (Extended Data Fig. 8d,e), unlike strains that evolved under cross-feeding. In addition, we observed that, for both partners, the mutations that became selected under conditions of cross-feeding confer a growth disadvantage when these mutants are directly competed against the ancestral strains in AH-MM (Extended Data Fig. 7g,h). Therefore, these mutations are unlikely to be selected in supplemented media. Finally, as mentioned above, the yeast *gdh1* mutation apparently collapses the community when this mutant is combined with the ancestral *E. coli* partner but it is selected in the co-culture with *E. coli argR* mutant. We have now modified the results and discussion to elaborate on these points, and we thank the Reviewer for these helpful suggestions.

Reviewer #2 (Remarks to the Author):

This manuscript presents a detailed set up of an interkingdom, simplified community made of a bacterium and a yeast, to study the evolution of the metabolic interdependency of nitrogen use. By studying the different generations of the two cross-feeding members of the co-culture, the study reports that several mutations in the two members of the community are selected to enhance the nutritional cooperation and interdependency. The study relies on a large set of interdisciplinary approaches from mutant constructions to genome analyses, proteomics, metabolomics and community manipulations providing a rich set of data. The study produces a large set of evidence that demonstrates the enhancement of the metabolic interdependency with a stronger metabolic “entanglement” between the two community components. It highlights the importance of “indirect selection” in reshaping such metabolic entanglement. The study is very interesting and well-designed providing very robust proofs of an enhanced cooperation. Two aspects should require attention: i) the level of novelty related to previous studies; ii) the connection between the introduction and the discussion of the results, regarding the robustness and rationale of the hypothesis at the base of the study. Regarding point i) It would be useful to read in the discussion an assessment of the result in comparison to previous studies that may help to highlight the novelty and importance of the study; also some discussion on the relevance of the findings for interactions in real world communities would enrich the manuscript. As per point ii) in the introduction the hypothesis proposed points out that interkingdom communities should evolve metabolic specialization more likely than communities of just bacteria or microbial eukaryotes. Besides the fact that a robust rationale on which this hypothesis is built is not clearly presented in the introduction, such hypothesis is not then developed in the discussion that touches subjects that are interesting, but not specifically related to the raised hypothesis.

We thank the reviewer for the appreciation of our work and for summarizing its key findings, and also for the constructive and helpful suggestions how to sharpen the presentation and discussion of our results. We have now substantially revised and extended the introduction to simplify the comparison to previous studies and to better justify the choice of the model system (which was partly described in the discussion section of the previous version of our manuscript). We believe that these changes have substantially improved the clarity of the presentation.

Specific comments

1. Lines 12-13: The information value of this sentence is questionable. The importance of metabolic interactions in the selection processes in general are very well demonstrated and, as well, their importance in the formation of complex life forms is self-evident. It is suggested to rephrase or remove the sentence. The following sentence slightly rephrased may be an incipit of the abstract that is clearer than the current one: "Different stages of mutualism can be observed in nature, however, the dynamics and mechanisms underlying.....".

As the reviewer suggested, this first sentence has been deleted and the subsequent sentence rephrased.

2. Lines 20-23: These lines are difficult to read, because they imply definitions not available in an abstract. The exact meaning of "indirect selection" is elusive in the context of the present text (especially in relation to pleiotropy).

This part of the abstract has been rephrased. We now better explain our definition of indirect selection in the abstract.

3. Line 38: "passive metabolic interactions" should be briefly explained/defined.

This sentence has been rephrased.

4. Line 48-52: From the context the current hypothesis is hardly to be considered as such, because it coincides with the observation presented at lines 45-46 that "the most pronounced examples of metabolism reduction are observed for symbiotic interactions between prokaryotic and eukaryotic partners". A hypothesis should be different from an observation that can support it. The new concept within a hypothesis should be generated by the assembly of different solid arguments. This is an essential aspect of the study, that is the hypothesis tested should be solidly justified. In the current presentation it is not indicated the rationale of why interkingdom microbial communities (prokaryote with eukaryote) should more likely undergo a metabolic specialization, than communities of solely bacteria or microbial eukaryotes.

We apologize for the unclarity in the formulation of the aims of our study and the underlying hypotheses. We now rephrased the introduction to hopefully better explain the rationale behind our work and the choice of the model system.

Our main rationale to hypothesize that this community might be more likely to undergo a metabolic specialization was not simply because it consists of a prokaryote and a eukaryote, but because of the differences in metabolic profiles of the two partners. This was previously mentioned in the discussion, but we have now moved this argument to the introduction to improve clarity. However, this was only one of the arguments for the choice of this model system rather than the hypothesis we aimed to test, as we hopefully make clearer now in the revised version of the introduction. The aim of our study was to reproduce, track and mechanistically explain the increase in the interdependence between partners during experimental evolution, which we indeed achieved.

5. Introduction: The previous comment highlights a general problem of the introduction that does not sufficiently explain the context and the rationale of the study (and the hypothesis).

As stated above, the introductory section has now been modified.

6. Extended Data Fig 1: What the reason of the choice of the co-cultures maintained for multiple (10-15) passages? If the growth is measured just with the OD, how can be measured the stability of auxotrophy?

We apologize for the missing piece of information. The shortlisted communities were selected on one hand to cover a broad range of final OD600 values after 120 h incubation, and on the other hand to test several different communities in which either one or both partners were interrupted in the same metabolic pathway but at different positions. We now explain it in the text and in the figure legend.

As for the second question, besides measuring OD, the final culture was assessed by isolating individual partners as described in the “Experimental evolution” section of Materials and methods and testing their growth as monoculture in the minimal media in absence of supplements. The figure legend has been modified to explain this.

7. Lines 62-63: The community combination indicated here is stated to be propagated stably but on panel a of extended data Fig 1 the OD seems very low.

In the previous version of our manuscript, the heatmap presented in Extended Data Fig.1a showed growth of the aggregating communities after 120 h of cultivation, at which time point this particular combination indeed reaches only low OD (as could also be seen in Fig. 1d). However, our selection was made primarily based on the non-aggregating community (previously shown in Extended Data Fig.1b), in which case the selected combination already reaches relatively high density after 120 h (see also Fig. 1b,c). To improve the clarity and avoid potential confusion, we have now exchanged these two panels of Extended Data Fig.1, showing the data for the non-aggregating community in panel a, and those for the aggregating community in panel b. We also explain it better in the text. We thank the Reviewer for raising this question.

8. Line 71 and Figure 1: This is not very clear and may confuse the reader. For instance, in Fig 1b S is the ancestral co-culture grown in AH-MM or is after 100 generations as it is apparent from the text in line 85? If this is the case the text and Fig 1b and the figure caption should be consistent, all indicating that S is after 100 generations in non-cross-feeding medium (AH-MM).

We apologize for the unclear description of this experiment. “S” indeed represent communities evolved for 100 generations in AH-MM (i.e., in supplemented medium). The plot represents the growth in CF-MM of the ancestral communities, communities evolved in CF-MM and communities evolved in AH-MM. The figure legend and the text have now been rephrased to clarify this point.

Another point: No differences are observed between 50 and 269 generations, raising the question when (around which generation) the growth changed?

As shown in Fig. 3, the main impact on community growth results from the inactivation of *argR* in *E. coli*, which happens quite early (in less than 7 generations). We now comment on it in the results section describing the order of mutation appearance and their phenotypic effects.

In the ancestral culture(s) apparently there are two growth phases (diauxia/crabtree effect?) that are not present in the CF (evolved) cultures. How have been considered any eventual effect of yeast-produced ethanol on *E. coli*?

Our preferred hypothesis to explain this apparent growth delay phase, and its shortening in the evolved communities, is that the delay is caused by the need to accumulate sufficient levels of arginine in the medium for cross-feeding. In support of this, the major growth improvement, including a near abolishment of the delay phase, could be recapitulated by co-culturing the *argR* mutant of *E. coli* (which secretes more arginine) with the ancestral yeast partner (Fig. 3a-c and Extended Data Fig. 7i). We now elaborate on that in the text.

In the figure caption panels (h,i) are g,h in the figure.

We sincerely apologize for the wrong labelling. The figure has now been corrected.

9. Line 76: What exactly mean “nearly proportional”?

The statement was indeed ambiguous and it has now been rephrased.

10. Figure 2 and lines 103-119: The text and figure here should benefit of a revision for clarity. One difficulty for the reader is to match the time and generations. In the figure panel a), time of mutation appearance is presented to illustrate the mutation, while in the text (e.g. line 116 for *E. coli*, but the same for the yeast lines reported later at lines 145-146 for example) the number of generations is used making difficult to follow where the number of generations as well the number of lines (“...4 out of 8 lines after 50 generations...” are presented.

We apologize for the unclarity of the presented data. The plot has now been changed to show the approximate number of generations at which the sequencing was performed and mutations were detected.

11. Line 111-112: Is this an original hypothesis presented here or it is an observation made previously for instance in reference 35?

This is indeed an assumption based on the previously reported repression of *E. coli* histidine transporter by ArgR. We rephrased the text to make it clear, and we thank the Reviewer for pointing out this ambiguity in our previous phrasing.

12. Lines 113-115: This exemplify well the concept of indirect selection that was introduced in the introductory section of the manuscript. For non-expert readers it would be useful to get such kind of clarification already in the introductory part.

We agree. As stated previously, the abstract has been modified to include the Reviewer's suggestions and to better explain the concept of indirect selection there.

13. Figure 4J: In this panel it should be useful that the meaning of the different type of arrows (thickness, dashed lines, colours, etc) may be explained in the caption.

We apologize for the incomplete description. The legend has now been adjusted accordingly.

14. Discussion: The current discussion is presenting several interesting aspects, but is not well linked to the introduction and does not provide a conceptual continuity to the hypothesis raised in the introductory part of the manuscript. It is suggested to revise the discussion/introduction to make them more coherent and with a direct evident conceptual linkage.

We thank the Reviewer for these suggestions. As already mentioned above, both the introduction and discussion have been modified to better introduce our hypotheses and aims and to improve the discussion of the results in the context of the literature.

Reviewer #3 (Remarks to the Author):

We thank the Reviewer for the fruitful comments that were included in one of the previous Reviewer's comments.

Reviewers' Comments:

Reviewer #1:

Remarks to the Author:

I thank the authors for addressing my comments. I am happy with their changes.

Reviewer #2:

None

Reviewer #3:

None

Point-by-point response to the reviewer comments

Reviewer #1 (Remarks to the Author):

I thank the authors for addressing my comments. I am happy with their changes.

We thank the Reviewer once again for the helpful comments, and we are happy that we could address them.